# Research on the Multiagent Joint Proximal Policy Optimization Algorithm Controlling Cooperative Fixed-Wing UAV Obstacle Avoidance

**DOI:** 10.3390/s20164546

**Published:** 2020-08-13

**Authors:** Weiwei Zhao, Hairong Chu, Xikui Miao, Lihong Guo, Honghai Shen, Chenhao Zhu, Feng Zhang, Dongxin Liang

**Affiliations:** 1Changchun Institute of Optics, Fine Mechanics and Physics, Chinese Academy of Sciences, No. 3888, Dongnanhu Rd., Changchun 130033, China; zhaoweiwei215@mails.ucas.ac.cn (W.Z.); guolh@ciomp.ac.cn (L.G.); zhuchenhao17@mails.ucas.ac.cn (C.Z.); 2University of Chinese Academy of Sciences, No. 19, Yuquan Rd., Beijing 100049, China; 3School of Information Engineering, Henan University of Science and Technology, Luoyang 471000, China; miaoxikui@gmail.com; 4Key Laboratory of Airborne Optical Imaging and Measurement, Changchun Institute of Optics, Fine Mechanics and Physics, Chinese Academy of Sciences, No. 3888, Dong Nanhu Road, Changchun 130033, China; shenhh@ciomp.ac.cn; 5School of Aviation Operations and Services, Aviation University of the Air Force, No. 2222, Dongnanhu Rd., Changchun 130022, China; zf00401@outlook.com; 6Xi’an Jiaotong University Health Science Center, No. 76, Yanta West Road, Xi’an 710061, China; abutfdevil@outlook.com

**Keywords:** reinforcement learning, proximal policy optimization (PPO), the joint state-value function, multiagent cooperative, multiple unmanned aerial vehicles (multi-UAV) formation, obstacle avoidance

## Abstract

Multiple unmanned aerial vehicle (UAV) collaboration has great potential. To increase the intelligence and environmental adaptability of multi-UAV control, we study the application of deep reinforcement learning algorithms in the field of multi-UAV cooperative control. Aiming at the problem of a non-stationary environment caused by the change of learning agent strategy in reinforcement learning in a multi-agent environment, the paper presents an improved multiagent reinforcement learning algorithm—the multiagent joint proximal policy optimization (MAJPPO) algorithm with the centralized learning and decentralized execution. This algorithm uses the moving window averaging method to make each agent obtain a centralized state value function, so that the agents can achieve better collaboration. The improved algorithm enhances the collaboration and increases the sum of reward values obtained by the multiagent system. To evaluate the performance of the algorithm, we use the MAJPPO algorithm to complete the task of multi-UAV formation and the crossing of multiple-obstacle environments. To simplify the control complexity of the UAV, we use the six-degree of freedom and 12-state equations of the dynamics model of the UAV with an attitude control loop. The experimental results show that the MAJPPO algorithm has better performance and better environmental adaptability.

## 1. Introduction

The autonomy and intelligent development of the coordinated control of multi-agent systems such as multi-unmanned aerial vehicle (UAV) and multi-robot have received more and more attention. To solve the problem of coordinated control and obstacle avoidance of multiagent systems, researchers have proposed various solutions, including rule-based methods, field methods, geometric methods, numerical optimization methods, and so on [1,2].

In recent years, the application and development of reinforcement learning (RL) in the field of robotics has attracted more and more attention. La et al. [3] integrated high-level behaviors by RL and low-level behaviors by flocking control to allow robots to learn to avoid predators/enemies collaboratively. Hung et al. [4] proposed a method of using the RL algorithm to fixed-wing UAV flocking, but the velocity and height of the UAV were set as constant to reduce the complexity of UAV control. Pham et al. [5] provided a framework for integrating Q-learning and Proportion Integration Differentiation (PID) to allow the UAV to navigate successfully. Koch et al. [6] investigated the performance and accuracy of the inner control loop, providing attitude control. Their investigations encompassed flight control systems trained with RL algorithms, including Deep Deterministic Gradient Policy (DDGP), Trust Region Policy Optimization (TRPO), and Proximal Policy Optimization (PPO) compared with PID controller. Fang Bin et al. studied the application of RL in the field of multi-UAV collaborative obstacle avoidance [2]. 

The success of reinforcement learning (RL) in the field of multiagent has been witnessed. However, it must also be pointed out that the development of multiagent reinforcement learning (MARL) is influenced by single-agent RL, which is the cornerstone of MARL development. There are three task environments in the multiagent field: cooperative tasks, competitive tasks, and mixed tasks [7]. We mainly consider the cooperative task environment. RL in the topic of multiagent cooperation is mainly used to optimize a common reward signal. The non-stationary environment due to the change of the policies of learning agents in a multiagent environment leads to the failure or difficulty of convergence of most single-agent RL algorithms in a multiagent environment [8,9]. Therefore, it is inevitable that we must find RL algorithms suitable for a multiagent environment.

Tan [10] explored the multiagent setting using independent Q-learning to complete some cooperative tasks in a 2D grid world. Tampuu et al. [11] manipulated the classical rewarding scheme of Pong to demonstrate how competitive and collaborative behaviors emerge by independent Q-learning (IQL). Foerster et al. [12] proposed a counterfactual multiagent (COMA) policy gradients algorithm by using a centralized critic to estimate the Q-function and decentralized actors to optimize the agents’ policies and evaluated on StarCraft. Lowe et al. [13] extended Deep Deterministic Policy Gradient (DDPG) to a multiagent setting with a centralized Q-function and evaluated 2D games. Li et al. [14] proposed a novel minimax learning objective based on the multiagent deep deterministic policy gradient algorithm for robust policy learning. Yang et al. [15] proposed the Mean Field Reinforcement Learning (MFRL) algorithm to address MARL on a very large population of agents. Value-Decomposition Networks (VDN) [16] learned a centralized action–value function as a sum of individual action–value functions and a decentralized policy. QMIX [17] is an effective improvement of the VDN algorithm but unlike VDN. QMIX can learn a complex centralized action–value function with a factored representation that scales well in the number of agents and allows decentralized policies to be easily extracted via linear-time individual argmax operations. Hong et al. [18] introduced a deep policy inference Q-network (DPIQN) and its enhanced version deep recurrent policy inference Q-network (DRPIQN) to employ “policy features” learned from observations of collaborators and opponents by inferring their policies. Bansal et al. [19] explored that a competitive multiagent environment trained with self-play can produce complex behaviors than the environment itself using Proximal Policy Optimization (PPO). There are also many review articles [7,8,9] that explore the research and application of RL in the field of multiagent.

In order to make RL algorithms useful in the real world, researchers have exerted a lot of effort [3,5,6]. The unmanned aerial vehicle (UAV) is a popular tool in the military and civilian fields, and it is also the representation of agents in the real world. The UAV, as the research object of RL algorithms, has attracted the attention of many researchers [5,6,20]. Multi-UAV formation can accomplish many tasks that cannot be completed by a single UAV [21,22].

This paper proposes the Multiagent Joint Proximal Policy Optimization (MAJPPO) algorithm, which uses the moving window average of the state–value functions of different agents to get the centralized state–value function to solve the problem of multi-UAV cooperative control. The algorithm can effectively improve the collaboration among agents in the multiagent system than the multiagent independent PPO (MAIPPO) algorithm. Since the PPO algorithm uses a state–value function as the evaluation function, it is different from Deep Q-Network (DQN) [23], which uses the action–value function as the evaluation function. Therefore, the centralization value function of the MAJPPO algorithm does not require the policies of collaborative agents during training, thereby reducing the complexity of the algorithm. Finally, we use algorithms to train multi-UAV formation through a multi-obstacle environment to evaluate the performance of the algorithm. In the process of reinforcement learning of UAV, there are two choices of the controlled object. One is the use of the UAV dynamics model with an attitude control loop, and the other is the use of the UAV dynamics model without an attitude control loop. We use the UAV dynamics model with the attitude control loop as the control object of multi-UAV cooperative control. This is mainly because the UAV dynamic model with the attitude control loop has less freedom and fewer optimization targets.

Therefore, the main contributions of the paper are as follows:1The development of the MAJPPO algorithm; and,2The MARL algorithm is applied to the multi-UAV formation and obstacle avoidance field.

In regards to the rest of the paper, we will firstly introduce the background related to this paper in Section 2. Section 3 describes the PPO algorithm. Section 4 presents the independent PPO algorithm for the multiagent environment. Section 5 describes the novel MAJPPO algorithm, and it brings in some discussion. Section 6 describes the dynamics model of small UAV with the attitude control loop, RL of a single UAV, and the basic settings of the formation. Section 7 introduces experiments and analysis. The conclusions appear in Section 8.

## 2. Background and Preliminary

In the field of RL, the Markov decision process (MDP) is a key concept. RL enables the agent to learn a policy with good profits through interaction with the environment in an unknown environment. Such environments are often formalized as Markov Decision Processes (MDPs), described by a five-tuple (*S; A; P; R; γ*). At each time step *t*, an agent interacting with the environment observes a state $s_{t}\in S$, and chooses an action $a_{t}\in A$, which determines the reward $r_{t}∼R\left(s_{t};a_{t}\right)$ and next state $s_{t+1}∼P\left(s_{t};a_{t}\right)$. The purpose of RL is to maximize the cumulative discount rewards $G_{t}=\underset{\tau =t,\;T}{\sum }\gamma^{\tau -t}r_{\tau }$, where *T* is the time step when an episode ends, *t* denotes the current time step, *γ* ∈ [0, 1] is the discount factor, and $r_{\tau }$ is the reward received at the time step *τ*. The action–value function (abbreviated as Q-function) of a given policy π is defined as the expected return starting from a state–action pair (s,a), expressed as $Q^{\pi }\left(s,a\right)=E[G_{t}|s_{t}=s,a_{t}=a,\pi ]$. Q-learning is a widely used RL algorithm. Q-learning mainly uses the action–value function $Q^{\pi }\left(s,a\right)=E[G_{t}|s_{t}=s,a_{t}=a,\pi ]$ to learn the policy [24]. DQN [23] is a kind of RL algorithm combining Q-learning and neural network, which learns the action–value function Q * corresponding to the optimal policy by minimizing the loss: $L\left(\theta \right)=E_{\pi }\left[{\left(Q_{\theta }\left(s,a\right)-y\right)}^{2}\right]$, where $y=r+\gamma maxQ_{\theta^{\prime }}\left(s^{\prime },a^{\prime }\right)$, and y represents the Q-learning target value.

In the real world, the agent often cannot obtain all the information of the environment, or the environment information obtained by the agent is incomplete and noisy, that is, only part of the environment can be observed. In this case, we can use the partially observable Markov decision process (POMDP) to model such problems. A POMDP can be described as a six-tuple 〈*S; A; P; R; γ; O*〉, where *O* is the observation perceived by the agent. The deep recurrent Q-network (DRQN) [25] is proposed to deal with partially observable problems and POMDP problems, which extends the architecture of DQN with Long Short-Term Memory (LSTM).

In a multiagent learning domain, the POMDP generalizes to a stochastic game or a Markov game.

A multiagent learning environment can be modeled as a decentralized POMDP (Dec-POMDP) framework [26]. The Dec-POMDP model extends single-agent POMDP models by considering joint actions and observations.

Solving the Dec-POMDP problem is at the core of the MARL algorithms. The mainstream solution is to optimize the decentralized policy by centralized learning, such as MADDPG, VDN, and QMIX.

## 3. PPO Algorithm

Policy gradient (PG) methods are the same as Q-learning in the sense that they explicitly learn a stochastic policy distribution $\pi_{\theta }$ parametrized by θ. The objective of PG is to maximize the expected return over the trajectories induced by the policy $\pi_{\theta }$. If we denote the reward of a trajectory τ generated by policy $\pi_{\theta }\left(\tau \right)$ as r(τ), the policy gradient estimator has the form $g≔E\left[r\left(\tau \right)\nabla_{\theta }\log\pi_{\theta }\left(\tau \right)\right]$. This is the REINFORCE algorithm [27]. However, the REINFORCE algorithm has a high variance. A baseline, such as a value function baseline, can be used to improve the shortcomings of this type of algorithm. A generalized advantage estimate [28] is to use this method at the expense of some bias to reduce variance.

Schulman et al. [29] proposed that the TRPO algorithm can solve the shortcomings of the PG method that needs to be carefully adjusted for the step size. The PPO algorithm [30,31] is a simplification of the TRPO algorithm, which has a simpler execution method and sampling method.

The PPO algorithm optimizes the surrogate objective (1):(1)$$L^{clip}\left(\theta \right)=E\left[\min\left(l_{t}\left(\theta \right){\hat{A}}_{t},clip\left(l_{t}\left(\theta \right),1-\epsilon ,1+\epsilon \right){\hat{A}}_{t}\right)\right]$$
where $l_{t}\left(\theta \right)=\frac{\pi_{\theta }(a_{t}|s_{t})}{\pi_{\theta_{old}}(a_{t}|s_{t})}$ denotes the likelihood ratio and ${\hat{A}}_{t}$ is the generalized advantage estimate.

Similar to the DRQN algorithm, the combination of PPO and LSTM has a good effect on solving the POMDP problem [19,31].

## 4. Multiagent Independent PPO Algorithm

Tampuu et al. [11] demonstrate how competitive and collaborative behaviors emerge by independent Q-learning. Bansal et al. [19] explored that a multiagent environment produces complex behaviors by independent Proximal Policy Optimization (PPO) algorithm (MAIPPO).

Of course, the MAIPPO algorithm may also cause policies to fail to converge due to environmental non-stationary caused by the changing policies of the learning agents. The network structure of MAIPPO is shown in Figure 1.

The structure of the MAIPPO algorithm we construct is relatively simple. Actor and critic networks of the MAIPPO algorithm are composed of LSTM layers and a series of fully-connected layers (abbreviated as FC layers). The critic network obtains state–value function $V\left(O_{t}\right)$ and optimizes the critic network by minimizing the loss. The generalized advantage estimate $A\left(O_{t},a_{t}\right)$ is calculated from the value function $V\left(O_{t}\right)$ and then to optimize the actor network by the surrogate objective.

To update the critic network by minimizing the loss function (2):(2)$$L_{t}=E_{t}\left[{\left(r_{t}+\gamma V\left(O_{t+1}\right)-V\left(O_{t}\right)\right)}^{2}\right],$$

To update the actor network by optimizing the surrogate objective (4):(3)$$L_{t}^{clip}\left(\theta \right)=E_{t}\left[\min\left(l_{t}\left(\theta \right){\hat{A}}_{t},clip\left(l_{t}\left(\theta \right),1-\epsilon ,1+\epsilon \right){\hat{A}}_{t}\right)\right],$$
(4)$$L_{t}^{clip+S}\left(\theta \right)=E_{t}\left[L_{t}^{clip}\left(\theta \right)+cS\left[\pi_{\theta }\right]\left(O_{t}\right)\right],$$
where *S* denotes an entropy bonus, and c is coefficient. We can use a truncated version of generalized advantage estimation (5), so:(5)$${\hat{A}}_{t}=\delta_{t}+\left(\gamma \lambda \right)\delta_{t+1}+\ldots +\ldots +{\left(\gamma \lambda \right)}^{T-t+1}\delta_{T-1},$$
(6)$$\delta_{t}=r_{t}+\gamma V\left(O_{t+1}\right)-V\left(O_{t}\right)$$

$O_{t}$ in the above formulas has the following expressions for agent *i* and agent *j*:(7)$$O_{t}=\{\begin{array}{c} O_{t}^{i}=\left(s_{t}^{i},s_{t}^{j,p}\right)\\ O_{t}^{j}=\left(s_{t}^{j},s_{t}^{i,p}\right) \end{array},$$
where $O_{t}^{i}$ represents the observation of agent *i*, which is the union of the state $s_{t}^{i}$ of agent *i* and the partial state $s_{t}^{j,p}$ of the state $s_{t}^{j}$ of agent *j*. $O_{t}^{j}$ is the same. According to the theory of RL, we know that agent *i* selects policy $\pi_{i}$ from $O_{t}^{i}$, and gets the reward and the next state $s_{t+1}^{i}$. That is to say, the policy $\pi_{i}$ of agent *i* has no direct influence on $s_{t+1}^{j,p}$.

In other words, $V^{i}$ obtained by the critic network of agent *i* evaluates of the partial state $s^{j,p}$ of agent *j*. However, the policy $\pi_{i}$ obtained by the actor network only affects the state $s^{i}$ of agent *i*. This creates a division between critic and actor, which is one of the directions in which various MARL algorithms strive to improve.

## 5. Multiagent Joint PPO Algorithm

In a multiagent learning environment, the environment becomes the non-stationary due to the changing policies of the learning agents. For the independent Q-learning algorithm, agents optimize policies through the local action–value function, which obstructs convergence.

There are a series of improved algorithms whose main purpose is to learn the centralized critic. The counterfactual multiagent (COMA) policy gradients algorithm and the multiagent Deep Deterministic Policy Gradient (MADDPG) use a centralized critic to estimate the Q-function and decentralized actors to optimize the agents’ policies. VDN is to optimize the decentralized policies by using the sum of Q-value functions of each agent as a centralized evaluation function. QMIX is an improved algorithm of VDN, which learns a more complex joint action–value function by constructing a mixed network. DPIQN and DRPIQN propose to employ policy features of collaborators and opponents to infer and predict their policies.

The MAJPPO algorithm is proposed based on the MAIPPO algorithm. Different from the Q-learning algorithm, which uses the action–value function to evaluate and optimize policy, the PPO algorithm mainly uses the state–value function and the generalized advantage estimate to evaluate and optimize a policy. The MAJPPO algorithm learns mostly to obtain the joint state–value function and the generalized advantage estimate to evaluate and optimize the distribution policies.

To enhance the stability of training and the cooperative between agents, we use the moving window average of the state–value functions of different agents to obtain joint state–value functions $V_{joint}^{i}$ (8) and $V_{joint}^{j}$ (9):(8)$$V_{joint}^{i}=\xi V^{i}\left(O_{t}^{i}\right)+\left(1-\xi \right)V^{j}\left(O_{t}^{j}\right),$$
(9)$$V_{joint}^{j}=\xi V^{j}\left(O_{t}^{j}\right)+\left(1-\xi \right)V^{i}\left(O_{t}^{i}\right),$$
where ξ is constant.

Agent *i* and agent *j* simultaneously obtain their respective observations $O^{i}$ and $O^{j}$, which include both observations of their own and partial observations of other agents. The state–value functions $V^{i}$ and $V^{j}$ are obtained through the processing of their respective critic networks. Then, to obtain the joint state–value functions $V_{joint}^{i}$ and $V_{joint}^{j}$ through the weighted average of the state–value functions. The joint state–value function $V_{joint}^{i}$ includes both the evaluation of the state of agent *i* and the evaluation of the state of other agents. The small (1 − ξ) in $V_{joint}^{i}$ is mainly to reduce the effect of the evaluation of the remaining state of the state $s^{j}$ of agent j except for $s^{j,p}$ on the joint state–value function. The state–value function $V_{joint}^{j}$ obtained by the agent *j* includes both the evaluation of the state of agent *j* and the evaluation of the partial state of other agents. The surrogate objective obtained by $V_{joint}^{i}$ and $V_{joint}^{j}$ optimize the actor networks to get the cooperative policy.

The value function that the MAJPPO algorithm learns through critic networks is a combination of state features with its states and other agents. The VDN’s paper pointers out that lazy agents arise due to the partial observability of state. The critic networks of the MAJPPO algorithm use global information to learn the value function. The advantage functions deriving from the value function are used to update actor networks. This can solve the lazy agent problem to some extent. The MAJPPO algorithm and VDN algorithm or QMIX algorithm have similarities. MAJPPO uses the weighted average of the state–value function of each agent to replace the local state–value function of each agent to achieve the goal of centralized learning.

## 6. Multi-UAV Formation

### 6.1. Dynamics Model of Small UAV and Attitude Control

We use the six-degree-of-freedom, 12-state equations of motion with the quasi-linear aerodynamic and propulsion models [32]. The model is provided in Appendix A. It is a fairly complicated set of 12 nonlinear, coupled, first-order, ordinary differential equations. Among the variables of these equations, in addition to 12 state variables $\left[p_{n};p_{e};h;u;v;w;\varphi ;\theta ;\psi ;p;q;r\right]$, there are four input variables: the aileron deflection is denoted by $\delta_{a}$, the elevator deflection is denoted by $\delta_{e}$, and the rudder deflection is denoted by $\delta_{r}$ and the throttle command $\delta_{t}$.

We could use the attitude control method of Appendix B to control the attitude of the above-mentioned UAV dynamics model.

### 6.2. RL of Single UAV

We use the above-mentioned UAV model as the control body of RL. The basis of multiagent RL for multi-UAV collaborative control is that RL can control the stable flight of a UAV.

We have two ways to control the UAV’s stable flight by using reinforcement learning. One is to reinforce learning to control the dynamic model of the UAV directly, and the other is to reinforce learning to control the dynamic model of the UAV through the attitude loop. For the first method, the details are as follows. We use the 12-state of UAV as the input of the neural network of PPO. The network output of PPO is $\left[\delta_{a};\delta_{e};\delta_{r};\delta_{t}\right]$, where $\left[\delta_{a};\delta_{e};\delta_{r}\right]\in \left[-1,\;1\right]$ and $\delta_{t}\in \left[0,1\right]$. The output $\left[\delta_{a};\delta_{e};\delta_{r};\delta_{t}\right]$ is applied to the UAV motion model to obtain the next states of UAV after 0.1 s, and this cycle, as shown in Figure 2. We opt to use 10 s, which is 100 steps as an episode. Additionally, every 10 episodes update the network. A reasonable reward function structure is necessary to learn a stable control model. To accomplish such a task, the UAV starts from the appropriate position and reaches another position $\left[p_{n}^{target};p_{e}^{target};h^{target}\right]$ in a stable attitude and a certain velocity $V^{target}$. Then, this reward function (11) can be constructed like this:(10)$$r_{navig}=\left|p_{n}-p_{n}^{target}\left|+\right|p_{e}-p_{e}^{target}\right|,\;\mathrm{with}$$
(11)$$R_{single}=-r_{navig}-\left|h-h^{target}\right|-\eta_{v}\left|V-V^{target}\right|-\left|\varphi \right|-\left|\theta \right|-\left|\psi \right|-\left|p\right|-\left|q\right|-\left|r\right|$$
where $\eta_{v}$ is constant, and V=norm([u,v,w]). $r_{navig}$ can be transformed according to specific tasks. In this way, the UAV can get a stable policy model to complete the task.

For the second method, the details are as follows. We use the 6-state of UAV (position and velocity) as the input of the neural network of PPO. The network output of PPO is $\left[\theta ;\phi ;\delta_{t}\right]$, where [θ;ϕ]∈[−0.5, 0.5] and $\delta_{t}\in \left[0,1\right]$. The output $\left[\theta ;\phi ;\delta_{t}\right]$ is applied to the UAV dynamics model with attitude control loop to obtain the next states of UAV after 0.5 s, and this cycle, as shown in Figure 3. In order to learn a stable control model, a reasonable reward function structure is necessary. To accomplish such a task: the UAV starts from the appropriate position and reaches another position $\left[p_{n}^{target};p_{e}^{target};h^{target}\right]$ in a stable attitude and a certain velocity $V^{target}$. Then, this reward function (13) can be constructed like this:(12)$$r_{navig}=\left|p_{n}-p_{n}^{target}\left|+\right|p_{e}-p_{e}^{target}\right|,$$
(13)$$R_{single}=-r_{navig}-\eta_{h}\left|h-h^{target}\right|-\eta_{v}\left|V-V^{target}\right|,$$

In this way, the UAV can obtain a stable policy model to complete the task.

Comparing the two methods, we can find that controlling the UAV with the attitude loop has fewer optimization targets, which can reduce the complexity of the UAV control and facilitate multi-UAV coordinated control.

### 6.3. Multi-UAV Formation

We use the MAIPPO and MAJPPO algorithms to solve multi-UAV collaborative control tasks. The control tasks we studied mainly constitute three UAV formations and obstacle avoidance.

The inputs to the MAIPPO and MAJPPO algorithms include their states, distance from obstacles and partial states of other UAVs. We found that using positions of other UAVs in experiments can result in more stable training results. For example, three UAVs are represented by UAV1, UAV2, and UAV3, respectively. Moreover, two obstacles are represented as obstacle1 and obstacle2. Then, the network input of UAV1 is $\left[p_{n}^{1};\;p_{e}^{1};\;h^{1};\;u^{1};\;v^{1};\;w^{1};\;p_{n}^{2};\;p_{e}^{2};\;h^{2};\;p_{n}^{3};\;p_{e}^{3};\;h^{3};\;dist^{11};dist^{12}\right]$, where $\left[p_{n}^{2};p_{e}^{2};h^{2}\right]$ is the position of UAV2, $\left[p_{n}^{3};p_{e}^{3};h^{3}\right]$ is the position of UAV3, and so on. $dist^{11}$ and $dist^{12}$ are the distance between UAV1 and obstacle1 and the distance between UAV1 and obstacle2. UAV has a detection distance $d^{detection}$ for obstacles, when $dist^{11}>d^{detection}$, $dist^{11}=d^{detection}$. The reward function (16) consists of three parts: (I) one to fly UAVs with a stable attitude and velocity, denoted by $R_{single}$, (ii) another part to coordinate UAVs’ flight while maintaining a certain formation distance, denoted by $R_{form}$, (iii) the third part to implement UAVs’ obstacle avoidance, as follows.
(14)$$\begin{array}{c} R_{form}=|norm\left(p_{n}^{12},p_{e}^{12},h^{12}\right)-d_{form}|-\left|norm\left(p_{n}^{13},p_{e}^{13},h^{13}\right)-d_{form}\right|-\\ |norm\left(p_{n}^{23},p_{e}^{23},h^{23}\right)-d_{form}|-\left|norm\left(p_{n}^{21},p_{e}^{21},h^{21}\right)-d_{form}\right|-\\ |norm\left(p_{n}^{31},p_{e}^{31},h^{31}\right)-d_{form}|-\left|norm\left(p_{n}^{32},p_{e}^{32},h^{32}\right)-d_{form}\right|, \end{array}$$
(15)$$R_{obstacle}^{1}=\{\begin{array}{l} 0,\;dist^{11}>d^{detection}\\ -\frac{d_{form}}{dist^{11}},\;0<dist^{11}\leq d^{detection}\\ -1000,\;0\geq dist^{11} \end{array}$$
(16)$$R=\frac{1}{Κ}\left[\alpha_{s}R_{single}+\alpha_{f}R_{form}+\alpha_{o}\left(R_{obstacle}^{1}+R_{obstacle}^{2}\right)\right],$$
where K, $\alpha_{s},\alpha_{f},\;and\;\alpha_{o}$ are constants, and $\alpha_{s}+\alpha_{f}+\alpha_{o}=1$, $d_{form}$ represent the safe distance of the formation, $norm\;\left(p_{n}^{12},p_{e}^{12},h^{12}\right)$ represents the distance of between UAV1 and UAV2 while $norm\;\left(p_{n}^{13},p_{e}^{13},h^{13}\right)$ represents the distance of between UAV1 and UAV3. $R_{obstacle}^{1}$ and $R_{obstacle}^{2}$ are the reward function of UAV for obstacle1 and obstacle2.

## 7. Experiments

### 7.1. Experimental Condition

#### 7.1.1. Network Settings

Critic network architectures first process the input using an LSTM layer with 128 hidden units, and then a fully connected linear layer with 128 hidden units followed by a TanH layer, and then a fully connected linear layer with 128 hidden units followed by a TanH layer.

The actor consists of two parts: a neural network and a normal distribution. The actor network has an LSTM layer with 128 hidden units, and then a fully connected linear layer with 128 hidden units followed by a TanH layer. The output of the network is the mean value of the normal distribution with covariance matrix *C* = 0.05 *I*, where *I* is the identity matrix [33]. The distribution generates actions. The output range of the angles [θ;ϕ] in the actor output is limited to [−0.5, 0.5], and the range of the throttle $\delta_{t}$ is limited to [0, 1]. Therefore, the mean value of the angle uses TanH as the activation function, and the mean of the throttle uses sigmoid as the activation function.

Due to the computational complexity of the UAV motion model, to shorten the training time, the use of multiple processes is inevitable.

#### 7.1.2. Parameter Settings

The learning rate of Adam is 0.0001. The clipping parameter ϵ=0.2, discounting factor γ=0.995 and generalized advantage estimate parameter λ=0.95. We use large batch sizes, which can improve the variance problem to some extent and help to explore. In each iteration, we collect 1000 samples or 20 episodes, and 50 steps as one episode, and perform 20 episodes of training in mini-batches consisting of 512 samples. We found *l*_2_ regularization with parameter 0.01 of the policy and value network parameters to be useful. The coefficient of the entropy is c=0.001. the parameters in the reward function are set to $K=100,\;\alpha_{s}=0.5,\;\alpha_{f}=0.4,\;\alpha_{o}=0.1,\;\eta_{h}=5,\;\eta_{v}=2$. The sampling time is set to ∆t=0.5.

### 7.2. Mission Environment

Assume that the three UAVs fly from the initial area to the target area at a certain speed and a stable attitude as required by the formation, and pass through the area with six obstacles. The following initial values are assumed to simplify the task environment:(1)$p_{e}^{1}$ and $p_{n}^{1}$ of UAV1 are uniformly distributed in the interval [0, 100] and [0, 100] respectively,(2)The initial values of $p_{e}^{2}$ and $p_{n}^{2}$ of UAV2 are uniformly distributed in the interval [−100, 0] and [0, 100] correspondingly, and(3)The initial values of $p_{e}^{3}$ and $p_{n}^{3}$ of UAV3 are uniformly distributed in the interval [−50, 50] and [−100, 0].

The initial values of $h^{1}$, $h^{2}$, and $h^{3}$ of UAV1, UAV2, and UAV3 are uniformly distributed in the interval [160, 240]. The initial values of velocity *u*, *v* and *w* of UAV1, UAV2 and UAV3 are uniformly distributed in the interval [10, 40], [1, 5] and [1, 5], the initial values of angle and angular velocity *φ*, *θ*, *ψ*, *p*, *q*, and *r* are uniformly distributed in the interval [−0.5, 0.5]. The target area is set to $p_{e}^{target}=0$, $p_{n}^{target}=1000$ and $h^{target}=200$. We assume that the safety distance between UAVs is $d_{form}=50$. The formation flight velocity is $V^{target}=40$. Six spherical obstacle centers with a radius of 20 are uniformly distributed in a range of obstacles [400, 700] × [−200, 200] × [150, 250] in a uniformly distributed manner. The maximum detection distance of UAV to obstacles is $d^{detection}=200$.

### 7.3. Experimental Comparison and Analysis

In order to compare the performance of the algorithm intuitively, we use the above parameters and task environment to perform experiments on the MAIPPO and MAJPPO algorithms until convergence, where the parameter in the MAJPPO algorithm is ξ=0.9. The learning curves of the algorithms are revealed in Figure 4. It should be noted that the reward in Figure 4 is the sum of the rewards of three UAVs. We performed 10,000 iterations for the MAIPPO algorithm and the MAJPPO algorithm. It can be clearly seen from Figure 4 that the MAJPPO algorithm demonstrates better performance than the MAIPPO algorithm in dealing with multi-UAV collaboration and obstacle avoidance problems. It can also be seen from Figure 4 that the learning curve of the MAIPPO algorithm is not stable after convergence, and the MAJPPO algorithm can get higher reward value and convergence more stable. Therefore, the MAJPPO algorithm can get better results than the MAIPPO algorithm when dealing with this Dec-POMDP environment. The training learning curve of the MAIPPO algorithm cannot converge well because of the instability of the environment.

Figure 5 shows trajectory curves, distance curves, altitude curves, velocity curves, and distance curves between UAVs and obstacles of UAVs after training using MAIPPO algorithm and MAJPPO algorithm. As can be seen from Figure 5, these three UAVs can fulfill the mission requirements well.

It can be seen from Figure 5 that the network trained by the MAJPPO algorithm performs better in the multi-obstacle environment for multi-UAV obstacle avoidance control. To specifically evaluate the performance of distance, altitude, and velocity of UAVs, we calculate the sum of first-order absolute center moment separately as (17), (18), and (19).

For *d*:(17)$$\{\begin{array}{c} {\bar{e}}_{d}\left(t\right)=\frac{1}{N}\overset{N}{\underset{i=1}{\sum }}\left|d_{i}\left(t\right)-d_{form}\right|\\ E_{d}=\overset{T}{\underset{t=0}{\sum }}{\bar{e}}_{d}\left(t\right) \end{array}$$

For *h*:(18)$$\{\begin{array}{c} {\bar{e}}_{h}\left(t\right)=\frac{1}{N}\overset{N}{\underset{i=1}{\sum }}\left|h_{i}\left(t\right)-h^{target}\right|\\ E_{h}=\overset{T}{\underset{t=0}{\sum }}{\bar{e}}_{h}\left(t\right) \end{array}$$

For *v*:(19)$$\{\begin{array}{c} {\bar{e}}_{v}\left(t\right)=\frac{1}{N}\overset{N}{\underset{i=1}{\sum }}\left|v_{i}\left(t\right)-V^{target}\right|\\ E_{v}=\overset{T}{\underset{t=0}{\sum }}{\bar{e}}_{v}\left(t\right) \end{array}$$

As can be seen from Table 1, $E_{d},\;E_{h},\;\mathrm{and}\;E_{v}$ of the MAJPPO algorithm improve by 42.72% ((5408.17–3097.77)/5408.17 × 100%), 39.66%, and 8.23% compared with the MAIPPO algorithm. This shows that MAJPPO algorithm has better performance.

### 7.4. Parameter Evaluation

Since the weighted average parameter ξ in the MAJPPO algorithm has a great influence on the performance of the algorithm, we discuss and analyze it. The learning curves of MAIPPO and MAJPPO for ξ = 0.8, 0.9, 0.99, and 0.999 are shown in Figure 6.

In the MAJPPO algorithm, when ξ = 1, it is actually the independent PPO algorithm. This can be seen from Figure 6, when the value of ξ is closer to 1. The performance of the algorithm will also show similar performance to the independent PPO algorithm, such as ξ = 0.999. However, the performance of the algorithm does not become better as the value of ξ becomes smaller. For example, when ξ = 0.8, the performance of the algorithm is not as good as ξ = 0.9.

## 8. Conclusions and Future Work

Based on the MAIPPO algorithm, we propose the MAJPPO algorithm that uses the moving window averaging of state-valued function to obtain a centralized state value function to deal with multiagent coordination problems. The MAJPPO algorithm is also a kind of centralized training and distributed execution algorithm. We also presented a new cooperative multi-UAV simulation environment, where multi-UAV work together to accomplish formation and obstacle avoidance. In order to accomplish this task, we use the dynamic model of the UAV with attitude control capability as the control object. It can be seen from the experimental comparison that the MAJPPO algorithm can better deal with the partial observability of the state in the multiagent system and obtain better experimental results.

The comparison of the MAJPPO algorithm with other multi-agent reinforcement learning algorithms, such as MADDPG, VDN, and QMIX, is left for future work.

## Figures and Tables

**Figure 1 sensors-20-04546-f001:**
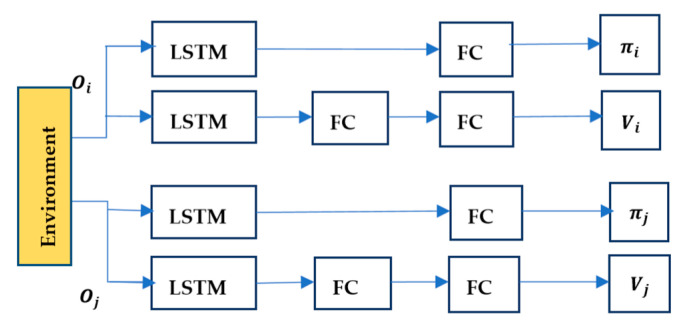
The network structure of the Multiagent Independent Proximal Policy Optimization (MAIPPO) algorithm.

**Figure 2 sensors-20-04546-f002:**
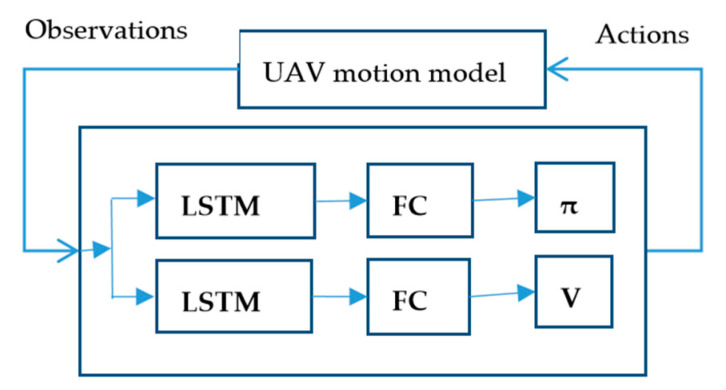
The single-agent PPO algorithm and the unmanned aerial vehicle (UAV) motion model for the feedback loop.

**Figure 3 sensors-20-04546-f003:**
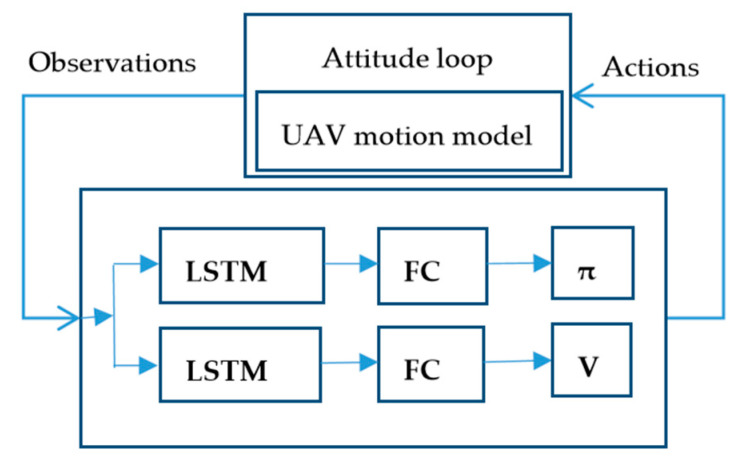
The single-agent PPO algorithm and the UAV motion model controlled by attitude loop for the feedback loop.

**Figure 4 sensors-20-04546-f004:**
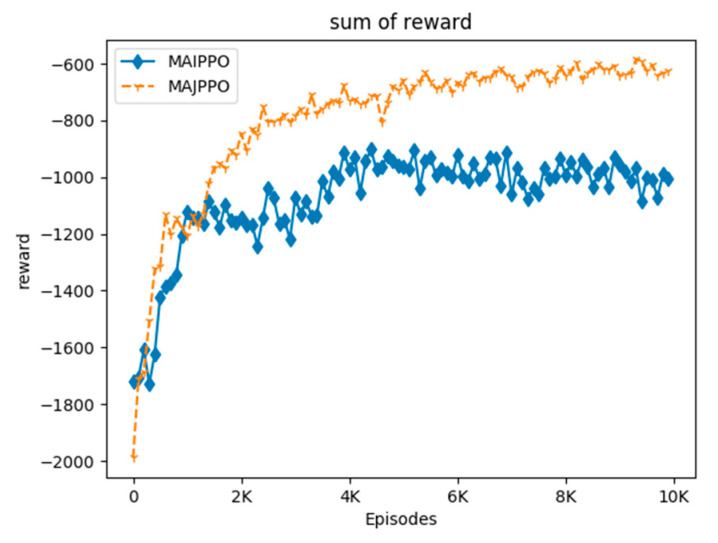
The curves of reward for multi-UAV formation uses the MAIPPO and the Multiagent Joint Proximal Policy Optimization (MAJPPO) for training.

**Figure 5 sensors-20-04546-f005:**
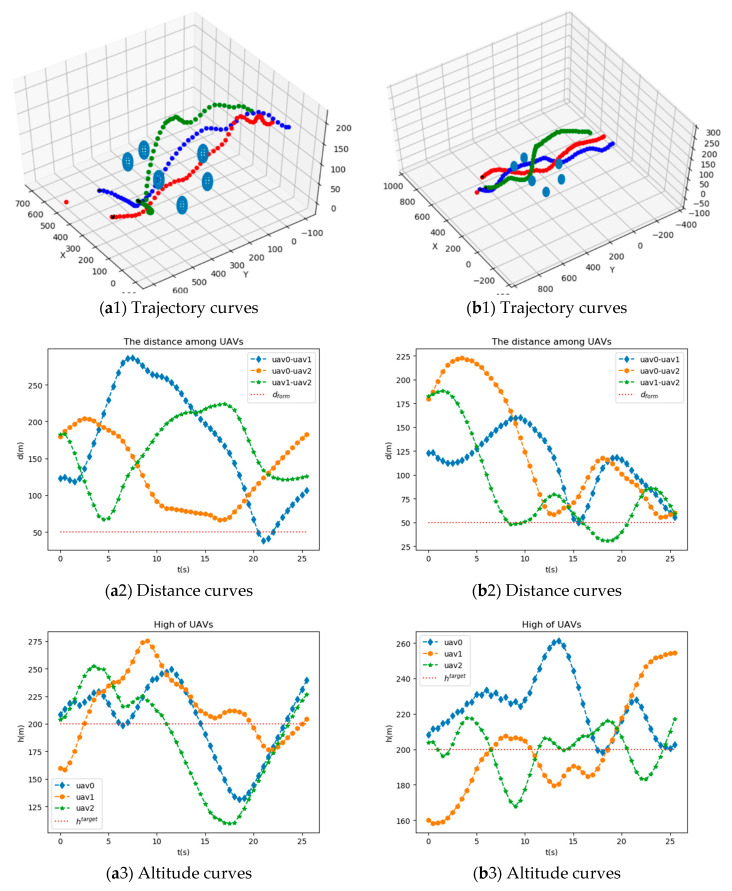
The trajectory curves, distance curves, altitude curves, velocity curves, and distance curves between UAVs and obstacles of UAVs after training using MAIPPO algorithm and MAJPPO algorithm. (**a**1–**a**5) is the result of the MAIPPO algorithm formation obstacle avoidance test, and (**b**1–**b**5) is the result of the MAJPPO algorithm formation obstacle avoidance test.

**Figure 6 sensors-20-04546-f006:**
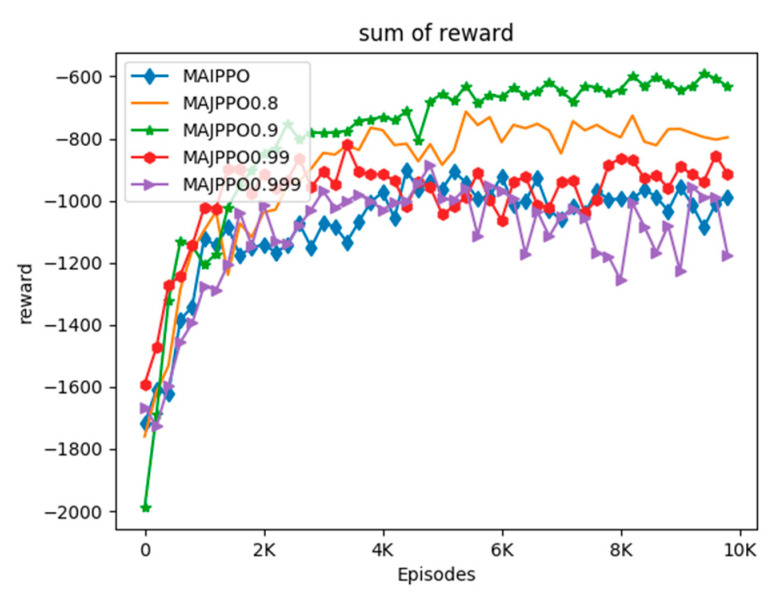
The learning curves of MAIPPO and MAJPPO for ξ = 0.8, 0.9, 0.99, and 0.999.

**Table 1 sensors-20-04546-t001:** The sum of first-order absolute center moment.

	Multiagent Joint Proximal Policy Optimization (MAIPPO)	Multiagent Independent Proximal Policy Optimization (MAJPPO)	Percentage
$E_{d}$	5408.17	3097.77	42.72%
$E_{h}$	1609.09	970.94	39.66%
$E_{v}$	126.46	116.05	8.23%

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
