# Peer review of "Research on the Multiagent Joint Proximal Policy Optimization Algorithm Controlling Cooperative Fixed-Wing UAV Obstacle Avoidance"

_sensors, 2020, doi:10.3390/s20164546_

Round 1

Reviewer 1 Report

The article entitled "Research on Obstacle Avoidance Method of Multi-UAV Formation Based on Multi-agent Reinforcement Learning Algorithms" is focused on the proposal and validation in simulated conditions of an improved multi-agent reinforcement learning algorithm, i.e., the multi-agent joint proximal policy optimization (MAJPPO) algorithm with the centralized learning and decentralized execution.

This issue is significant in the autonomous navigation of the swarm or fleet of unmanned aerial vehicles/unmanned aerial systems/remotely piloted aircraft systems (UAVs/UASs/RPASs), primarily in the context of avoiding air obstacles in the wide range applications, e.g., military, geomatics, 3D mapping and simultaneous localization and mapping (SLAM), monitoring of crops, land cover and land use, etc.

The article contains serious deficiencies (in order of appearance):

  1. The title of the article is imprecise. The title suggests comprehensive research of the proposed method, both in the aspect of a simulation study and during a practical experiment. In addition, the word "multi" appears twice in the title, which is not stylistically correct. I suggest that the authors consider using in the future the term "UAV swarm" instead of the "multi-UAV formation."
  2. The abstract of the paper is too general and not informative enough. The abstract should contain the only key thesis, research methodology, used solutions, and study results. The authors can extend the summary by approx. 50 words.
  3. The information contained in the introduction is not satisfactory. The introduction should better describe the main aim of the work. Justify practical needs, describe the background, and clearly define all research questions. Without the formal research question/hypothesis, it is difficult for the reader to understand, what are the real aims of the paper. Description of the content of individual chapters in the paper introduction is superfluous.
  4. The authors did not apply the MAJPPO algorithm performance test scenario for a large number of UAVs. The scenario of the presented experiment concerned flight simulation for 3 UAVs, i.e., the boundary condition for the term "multi." The authors also did not perform a practical experiment, at least for 3 UAVs, to verify and check the effectiveness of the proposed MAJPPO algorithm in real conditions flight of UAVs swarm. I believe the performed study is incomplete. Discussion of the results is also insufficient, because "the findings and their implications should be discussed in the broadest context possible and limitations of the work highlighted."
  5. The section "Conclusions" is too general formulated. The presented results are too brief and laconic. Conclusions should include precise, concise, and quantitative statements about the significance of the study, highlight any new findings. The aspect of future work is not described in this chapter. Acronyms should not be used in the "Conclusions." If used, they should be defined.
  6. The editorial quality of the article is unsatisfactory. In the article, the authors have inserted references to all figures and described all the variables present in the formulas; however, the mixed graphs (Figures: 3, 4, 5) are unreadable. They should be corrected. The scale and resolution of their presentation (editing) should be increased. Besides, all the presented equations (No. 1÷14) have no reference in the text of the paper. There is a mistake in the numbering of equations 10 and 11 (double numbering).

I appreciate the efforts done, but the scientific and editorial quality does not reach, in my humble opinion, the current version of the manuscript to accept for publication. I do not see the completeness of the performed research by the authors and their significant contribution to the development of science.

The authors are encouraged to consider the above comments and suggestions and revise the manuscript accordingly. After completing and correcting the content, the article can represent the appropriate scientific level. It may present an interesting study on utilitarian capability in the field of UAV swarm navigation based on multi-agent reinforcement learning algorithms.

Kind regards

Author Response

Revision of comments::

1、Research on the Multiagent Joint Proximal Policy Optimization Algorithm Controlling Cooperative Fixed-wing UAV Obstacle Avoidance

2、Thank you very much for your suggestions. We have modified the abstract and added the content of the abstract.

3、Thanks for your comment. The main purpose of the introduction is to illustrate that algorithms similar to MADDPG, VDN and other algorithms obtain a centralized value function. Our algorithm also optimizes distributed strategies by obtaining a centralized value function.

4、If a large number of UAVs are used for testing, the required time cost will be relatively large, and the reinforcement learning algorithm of the centralized value function is not conducive to the training and testing of a large number of agents. The actual test experiment of reinforcement learning is also our next step. In order to discuss the results in more detail, we did a closer test and comparison of the trained network, and got the trajectory curve, speed curve, height curve, etc.

5、We have revised the conclusion part and added an introduction to future work.

6、We have modified the diagram and formula.

Reviewer 2 Report

1) The enclosed file contains improvements to the original text as far as English and grammar go in YELLOW.

2) Since the authors actually introduce a new method, the title should reflect this contribution.

3) The text in PINK

"learning to indirectly control the dynamic model of the UAV through the attitude loop"

could become much better if you draw a diagram illustrating this. It is difficult to read.

4) Is it possible to talk about FANETs in the text? I ask that because realistically speaking the swarm will face some other problems during collision avoidance that will affect their communication. Hence, you have to keep control, communication, and energy needs in mind to decide effectively. Some topology changes may be necessary in view of these needs.

5) How does your fleet handle the failure of one flying node (UAV)? Can you link this to conditions of partial observability (and sensing within the fleet)? 

6) Can you talk about future improvements in the Conclusion?

7) Please, check your references and see if they adhere to the MDPI standards. there are several "et al." in the bibliography.

Author Response

Revision of comments:

1)Thank you very much for your suggestions. We have corrected grammatical errors.

2)We have revised the title.

3)The explanatory diagram has been drawn.

4)This kind of problem belongs to the delayed reward problem in reinforcement learning, and the delayed reward problem is more difficult to solve in reinforcement learning.

5)This question is similar to the previous question. If there is a node failure, the reinforcement learning algorithm may fail.

6)A description of future work has been added to the conclusion.

7)The references have been revised.

Reviewer 3 Report

In this paper, the authors addressed the design of a control strategy for UAV Formation based on reinforcement learning algorithms.

I found the manuscript difficult to read and difficult to understand. It seems to have some interesting results, but, in my opinion, the authors do not make them clear. I have the following comments for the authors further improvement:

  1. The main problem should be well defined and the paper title should be according to (e.g. UAV fixed-wing UAV ).
  2. The bibliographical review is extensive but shallow (Most part of the literature is related to RL). Authors should describe better the cited works and how they are related to the manuscript, allowing the reader to have a clear view of the current state of the field and of how the manuscript pushes it forward. There is a lack of reference about multi-agent formation reported by the control community, for instance, the seminal work [1].
  3. The paper needs to be rewritten to remove some redundant parts, making it more clear and concise. It seems very complicated now, and increase reading difficulty.
  4. The conducted simulations are very modest and don't really validate the author's claims.
  5. I understand the inherent difficulty in comparing results. Nevertheless, I suggest the authors offer a more thorough comparison, w.r.t. reported works to clarify their contribution.

[1] Olfati-Saber, ‘‘Flocking for multi-agent dynamic systems: Algorithms and theory,’’IEEE Trans. Autom. Control, vol. 51, no. 3, pp. 401–420,Mar. 2006.

Author Response

Revision of comments:

(1)Thank you very much for your suggestions. The title of the thesis has been revised.

(2)Our work is more about constructing a new type of multi-agent reinforcement learning algorithm and testing it in a multi-UAV formation obstacle avoidance environment to evaluate the algorithm.

(3) The content of the paper has been partially revised

(4) We have added some analysis.

(5) We have added the acknowledgements part.

Round 2

Reviewer 3 Report

The authors have made an enormous effort to improve their paper, and my comments have been taken into account. 

Please add the following references which were suggested in my first review:

‘‘Flocking for multi-agent dynamic systems: Algorithms and theory,’’ IEEE Trans. Autom. Control, vol. 51, no. 3, pp. 401–420, Mar. 2006.

"Leader-Following Consensus and Formation Control of VTOL-UAVs with Event-Triggered Communications", Sensors 2019, 19, 5498.

Author Response

Thank you very much for your suggestions. We have added references.

This manuscript is a resubmission of an earlier submission. The following is a list of the peer review reports and author responses from that submission.